# Learning to integrate parts for whole through correlated neural variability

**Zhichao Zhu**[1,2], **Yang Qi**[1,2,3]*, **Wenlian Lu**[4,5,6,7], **Jianfeng Feng**[1,2,3,8]*

**1** Institute of Science and Technology for Brain-Inspired Intelligence, Fudan University, Shanghai, China, **2** Key Laboratory of Computational Neuroscience and Brain-Inspired Intelligence (Fudan University), Ministry of Education, Shanghai, China, **3** MOE Frontiers Center for Brain Science, Fudan University, Shanghai, China, **4** School of Mathematical Sciences, Fudan University, Shanghai, China, **5** Shanghai Center for Mathematical Sciences, Shanghai, China, **6** Shanghai Key Laboratory for Contemporary Applied Mathematics, Shanghai, China, **7** Key Laboratory of Mathematics for Nonlinear Science, Shanghai, China, **8** Zhangjiang Fudan International Innovation Center, Shanghai, China

* yang_qi@fudan.edu.cn (YQ); jffeng@fudan.edu.cn (JF)

**Data Availability Statement:** The Caltech-UCSD Birds-200-2011 dataset and the pre-trained VGG16 model are publicly available from https://www.

## Abstract

Neural activity in the cortex exhibits a wide range of firing variability and rich correlation structures. Studies on neural coding indicate that correlated neural variability can influence the quality of neural codes, either beneficially or adversely. However, the mechanisms by which correlated neural variability is transformed and processed across neural populations to achieve meaningful computation remain largely unclear. Here we propose a theory of covariance computation with spiking neurons which offers a unifying perspective on neural representation and computation with correlated noise. We employ a recently proposed computational framework known as the moment neural network to resolve the nonlinear coupling of correlated neural variability with a task-driven approach to constructing neural network models for performing covariance-based perceptual tasks. In particular, we demonstrate how perceptual information initially encoded entirely within the covariance of upstream neurons' spiking activity can be passed, in a near-lossless manner, to the mean firing rate of downstream neurons, which in turn can be used to inform inference. The proposed theory of covariance computation addresses an important question of how the brain extracts perceptual information from noisy sensory stimuli to generate a stable perceptual whole and indicates a more direct role that correlated variability plays in cortical information processing.

## Author summary

Understanding how the brain represents and processes perceptual information through neuronal firing patterns is at the heart of neuroscience. The prevailing idea suggests that the information is primarily encoded in mean firing rates, whereas correlations among neurons may play a secondary role. However, given that firing variability is ubiquitously observed in cortical neurons, one wonders if correlated noise may play a more central role in neural computation than previously thought. Here, we propose that perceptual information can be encoded in part or even entirely in the correlated variability of spiking neurons. Through a combination of theoretical modeling and machine learning approaches,

vision.caltech.edu/datasets/cub_200_2011/ and
https://pytorch.org/vision/stable/models/vgg.html.
The code used in this article is available at https://
github.com/BrainsoupFactory/moment-neural-
network.

**Funding:** W.L. received funding from the National
Science and Technology Major Project of China
(No. 2018AAA0100303) and the National Natural
Science Foundation of China (No. 62072111). Y.Q.
received funding from the National Natural Science
Foundation of China (No. 62306078). This work is
also jointly supported by ZJ Lab and Shanghai
Center for Brain Science and Brain-Inspired
Technology. The funders had no role in study
design, data collection and analysis, decision to
publish, or preparation of the manuscript.

**Competing interests:** The authors have declared
that no competing interests exist.

we construct neural network models capable of processing correlated variability in a task-
driven way. We demonstrate that the trained network is able to learn to extract covari-
ance-encoded perceptual information to generate stimulus-selectivity in their mean firing
rates, thanks to the nonlinear coupling of statistical moments of their activity. Informa-
tion-theoretic analysis reveals a near-lossless transfer of perceptual information from the
covariance of upstream neurons to the mean firing rate of downstream neurons. Our
work offers new insights into the role of correlated variability in cortical processing and
hints towards a task-driven paradigm for studying cortical computation with biologically
plausible neural network models.

## Introduction

The firing rate of neural activity, defined as the number of spikes within a specific time win-
dow, is considered the primary carrier of information in the brain [1–3]. Cortical processing
under this rate code is relatively well understood. Starting with the groundbreaking research
by Hubel and Wiesel [4], which demonstrated that the firing rates of numerous neurons in the
primary visual cortex (V1) are systematically influenced by the retinal position and orientation
of visually presented edges in cats, and introduced a feedforward model to explain this mecha-
nism, a series of subsequent studies have utilized artificial nodes to explore similar information
processing questions [5, 6]. It is generally accepted that layers of neural networks form a fea-
ture hierarchy with a growing level of abstraction [7]. In this way, the brain can perceive the
world by integrating a collection of sensory signals that reflect the physical attributes of the
external world into a singular perceptual whole [8–10].

However, the environment in which humans and animals live is noisy and the sensory
information received is often limited or ambiguous. For example, due to the turbulent nature
of the air, odor concentrations can fluctuate significantly. Not only are stimuli noisy, but bio-
logical neurons also communicate through discrete spiking activities [11–13] which are tem-
porally irregular but consistent in amplitude. Moreover, the fluctuations in cortical activity
can exhibit rich correlation structures. These observations lead to the idea that neural compu-
tation is fundamentally probabilistic [14–16]. Despite the abundance of intrinsic and extrinsic
noise sources in the brain, our perception of the external world remains relatively stable. This
raises the question how the brain represents and processes noisy stimuli to generate a stable
perceptual whole.

Neural coding theories propose that the noise correlation structure can greatly influence
the effectiveness of a neural code, either by enhancing it through synergy or diminishing it
through redundancy [17–19]. A growing body of experimental and theoretical works indicate
the importance of correlated neural variability in neural population code [20–25]. In particu-
lar, through a synergy-redundancy mechanism, neural correlations can enhance or degrade
the information encoded by a neural population compared to independent neuronal
responses. Despite these advances, how correlated neural variability is propagated across neu-
ral populations and how they are involved in the processing of feature hierarchies are poorly
understood. There are two main challenges faced by this problem: one is the nonlinear cou-
pling between signal and noise of neural activity in biologically plausible spiking neural net-
works (SNN) [26] and the other is the construction of a neural network model that can
perform useful computations with correlated variability. Recent advances in theoretical model-
ing of spiking neural dynamics and machine learning have opened up new opportunities to
tackle these problems.

One of the advances from theoretical modeling studies is a computational framework known as the moment neural network (MNN), which naturally generalizes rate-based Wilson-Cowan models to second order [27–31]. Derived from first principles based on a Fokker-Planck formalism, the MNN accurately captures the statistical properties and nonlinear coupling of the correlated variability of spiking neural networks [27, 28]. An efficient numerical method has been developed allowing for rapid evaluations of the moment mappings without solving the underlying Fokker-Planck equation [29]. The main advantage of the MNN is that it faithfully describes the firing statistics of spiking neural networks while retaining the analytical tractability of continuous rate models.

With recent advances in deep learning [32], there is an increasing trend to use machine learning approaches to build neural network models to understand neural computation in the brain [33–35]. It has been shown that biologically plausible neural representations can emerge in trained network models with high representational similarity to neural responses in the brain [36]. An insight shared between experimental findings [8, 37] and machine learning studies [33–35] is that perceptual information grows more linearly separable during information processing, leading to the perceptual disentangling hypothesis [38, 39]. A number of studies have explored gradient-based learning with correlated variability, including a covariance perceptron [40] and its application in reservoir computers for classifying correlated time series data [41, 42]. More recently, it has been shown that standard deep learning architectures with rate-based neural networks can be systematically generalized to second-order statistical moments, allowing for learning in spiking neural networks with correlated neural variability [30]. These developments provide new opportunities for understanding neural computation with correlated neural variability in a task-driven way.

In this work, we introduce a novel theoretical framework of covariance-based computation with spiking neurons as a mechanism for perceptual inference. This computation is akin to a 'decorrelation' process [38, 39], transferring pertinent information from neuronal coactivities to the mean firing rate of individual neurons, thus facilitating the representation and downstream processing of perceptual information. We start by introducing an encoding scheme of sensory stimulus that partitions sensory neurons' covariance into components due to variations in the instantaneous firing rate and neuronal activity fluctuations, and then apply this encoding scheme to the motion direction of a moving grating. To implement covariance-based computation, we train the MNN to perform the task and show that the trained MNN can recover direction information with hidden layer neurons naturally exhibiting direction selectivity in their mean firing rate similar to cortical neurons [43, 44], without additional constraints. Spiking neural network simulations further demonstrate that covariance-based computation can be achieved by local fluctuations of spike trains, bypassing the need for explicit representation of the global covariance matrix. Information-theoretic analysis verifies a near-lossless recovery of pertinent information about the stimulus feature. We also reveal how covariance-based computation can be used to extract useful information from the rich covariance structure hidden within the feature maps of natural images and to facilitate inference by downstream neural populations. Our results challenge the traditional view that correlated variability is only a secondary factor in neural coding and highlight a more direct role that correlated variability plays in neural computation.

## Results

### Covariance-based neural coding and computational mechanisms

To elucidate the concept of covariance-based neural computation, let us consider a binary decision-making task involving the identification of a stimulus $s$ composed of two odor

concentrations $c_1$ and $c_2$ [45]. The co-release of these odors can be correlated or anticorrelated, contingent on the specific type of stimulus presented ($s = 1$ or $s = -1$). In this scenario, an animal needs to learn to discriminate between these two types of stimuli [45]. Although the average concentrations $c_1$ and $c_2$ of the odors are constant regardless of the stimulus type $s$, the actual concentrations that the animal's sensory neurons detect vary over time because of air turbulence. Consequently, the animal must decipher the temporal correlation patterns of these fluctuating odor concentrations to accurately differentiate between the two stimuli.

We can conceptualize the stimulus reaching the sensory neurons as being drawn from a structured distribution, as depicted in Fig 1a. This dynamic stimulus induces irregular spiking activities in sensory neurons. The downstream neurons are then tasked with interpreting the embedded information from these sensory neuron activity patterns. This scenario prompts an inquiry: What aspects of sensory neurons' activity convey the information about stimulus $s$, which is not captured by the mean firing rate?

We posit that information is naturally encoded through the covariance of temporally fluctuating stimuli. To illustrate the encoding of stimulus $s$, we introduce two different types of sensory neurons, each tuned to the component $c_1$ or $c_2$ of the stimulus, as shown in Fig 1b. The responses of these neurons are modeled as independent inhomogeneous Poisson processes with instantaneous firing rate $f(c_i)$, where $f$ is some function of the concentration $c_i(t)$ of odor $i$ at time $t$. Dividing time into non-overlapping bins gives a piecewise constant representation of

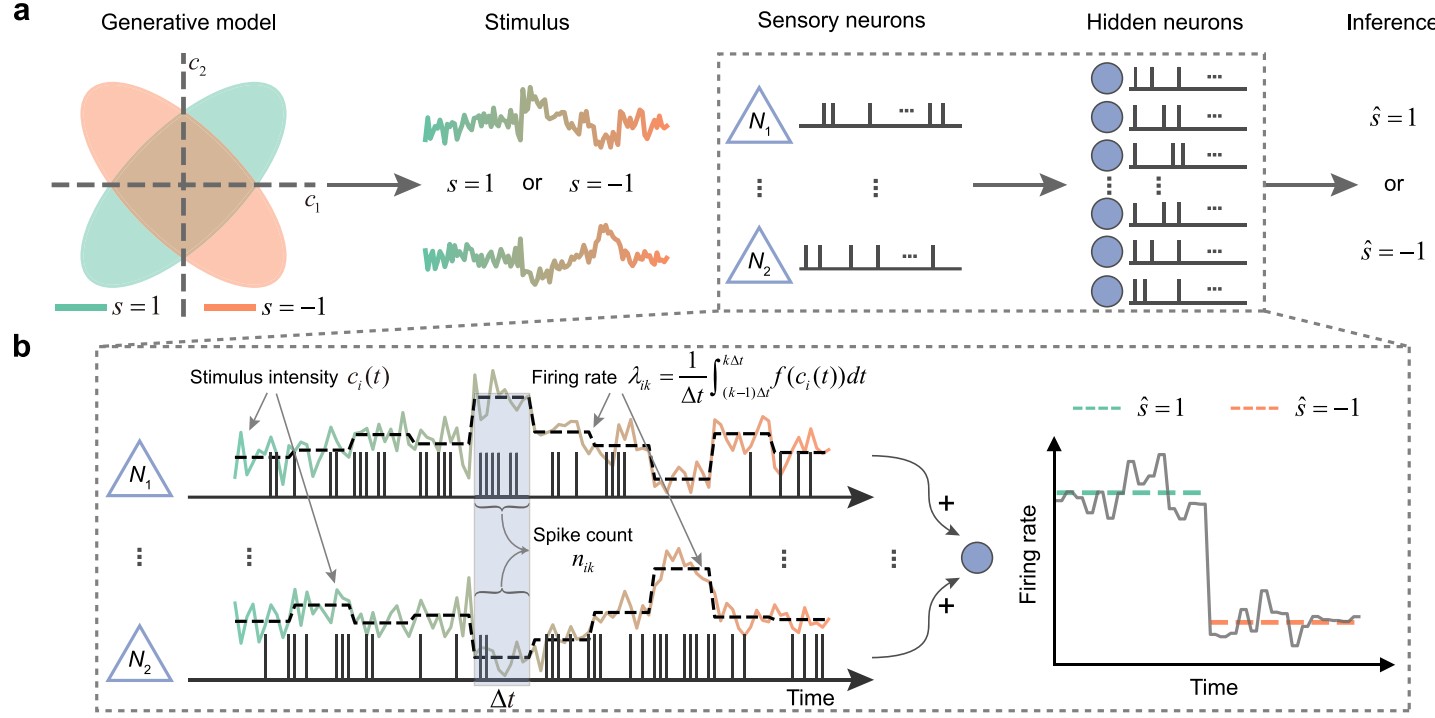

**Fig 1. Encoding perceptual information through temporal covariance in sensory neurons. a**, A schematic illustration depicting how a stimulus $s$ is composed of two components that can be either correlated or anticorrelated. The intensities of components $c_1$ and $c_2$ reaching the sensory neurons vary over time, resulting in fluctuating neural responses, from which the hidden neurons must infer $\hat{s}$. **b**, A detailed depiction of representative examples of the sensory and hidden neurons. The left panel shows how the varying firing rates of two representative sensory neurons reflect the intensity changes of the stimulus components. In the example, sensory neurons $N_1$ and $N_2$ respond to $c_1$ and $c_2$ respectively, such that the information about stimulus $s$ is encoded in the sign of their spike count correlation. The right panel shows how a hidden neuron can differentiate the two stimulus types by responding differently in its firing rate.

the mean firing rate within each bin $k$

$$\lambda_i(t) \doteq \lambda_{ik} = \frac{1}{\Delta t} \int_{(k-1)\Delta t}^{k\Delta t} f(c_i(t))dt, \quad (k-1)\Delta t \le t < k\Delta t, \tag{1}$$

where $\Delta t$ is a short time window over which sensory neurons can track changes [46, 47].

For a given time interval $\Delta t$, both the mean and variance of the spike count $n_{ik}$ are equal to $\lambda_{ik}\Delta t$. Denoting the expected value of the spike count conditioned on the firing rate as $\mathbb{E}[\cdot]$ and the average firing rate over time index $k$ as $\langle \cdot \rangle$, we can express the moments of spiking activity for a pair of neurons as

$$\mu_i \doteq \frac{1}{\Delta t} \langle \mathbb{E}[n_{ik}] \rangle = \langle \lambda_{ik} \rangle, \tag{2}$$

$$\Sigma_{ij} \doteq \frac{1}{\Delta t} \langle \mathbb{E}[(n_{ik} - \mu_i\Delta t)(n_{jk} - \mu_j\Delta t)] \rangle = \Sigma_{ij}^{\text{noise}} + \Sigma_{ij}^{\text{signal}}. \tag{3}$$

The first term on the right of Eq (3) is the noise covariance averaged over $k$. With the assumption that $n_{ik}$ represent independent Poisson spike count, this simplifies into

$$\Sigma_{ij}^{\text{noise}} = \delta_{ij}\mu_i, \tag{4}$$

where $\delta_{ij}$ represents the Kronecker delta. The second term corresponds to the cross-time signal covariance,

$$\Sigma_{ij}^{\text{signal}} = \langle (\lambda_{ik} - \mu_i)(\lambda_{jk} - \mu_j) \rangle \Delta t. \tag{5}$$

Note that both the mean firing rate and noise covariance, defined in Eqs (2) and (4) do not depend on the size of the time window chosen. In contrast, signal covariance is sensitive to the size of time window and diminishes when the stimulus is static ($\lambda_{ik} = \mu_i$) or when the observation window $\Delta t \to \infty$.

When the average intensity of the two stimulus components is the same, the perceptual information about $s$ cannot be discerned by observing $c_1$ or $c_2$ in isolation, but becomes evident when considering the correlation between these two components, as reflected by the correlated neural variability in sensory neurons. To illustrate how downstream neurons can infer stimulus $s$ from the correlated variability of sensory neurons, consider a hidden neuron connected to two sensory neurons with equal synaptic weights $w > 0$. In general, the mean firing rate of a spiking neuron receiving noisy inputs depends on both the input mean and the input variance. Therefore, the mean firing rate of the downstream neuron can be written as

$$\mu'(s) = \phi[w(\mu_1 + \mu_2), w^2(\sigma_1^2 + \sigma_2^2 + 2\sigma_1\sigma_2\rho(s))], \tag{6}$$

where $\rho(s)$ is the stimulus-dependent correlation coefficient between sensory neurons' responses, and $\phi$ is some neuronal activation function whose specific form depends on the type of spiking neuron (see Methods for details). In the scenario considered here, we have $\rho > 0$ when $s = 1$ and $\rho < 0$ when $s = -1$, therefore the total input variance for $s = 1$ is greater than that for $s = -1$. This difference is in turn reflected in the mean firing rate of the downstream neuron $\mu'(s)$ (right panel in Fig 1b), allowing the discrimination of the two stimulus types. In contrast, without the information provided by $\rho$ (such as when the sensory neurons fire independently), the mean firing rate of the downstream neuron would become insensitive to the stimulus type. Our analysis shows that, due to the nonlinear coupling between mean firing rate and firing variability, perceptual information initially encoded in the covariance $\Sigma$ can

be passed to the firing rates $\mu'$ of downstream neurons, thereby facilitating its interpretation and guiding decision-making.

## Representing motion directions through correlated neural variability

Having established the basic ideas of covariance-based computation, we now turn to demonstrate it in action with a motion direction detection task, commonly used to study early visual information processing. The task involves showing a subject a moving visual grating, oriented perpendicularly to its motion direction (illustrated in Fig 2a, leftmost panel), and having them identify its movement direction. The goal is to construct a feedforward neural network model that can infer the motion direction of a moving grating by exploiting its correlation structure. The first step towards this goal is to demonstrate how motion direction can be represented by the temporal correlations of the sensory neurons' responses to basic physical properties such as light intensity and its rate of change.

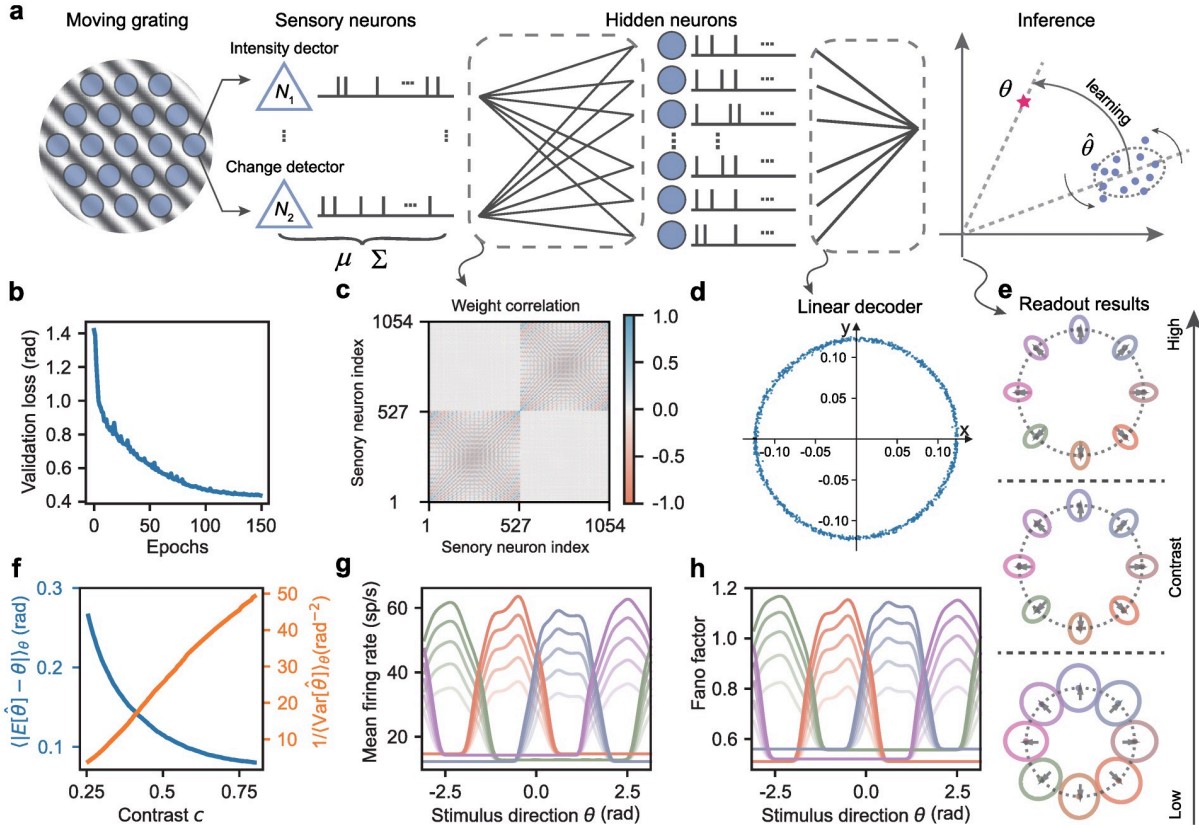

**Fig 2. Learning to infer motion direction through correlated neural variability. a,** Task overview: Sensory neurons organized in a hexagonal grid collectively respond to light intensity and its rate of change of a moving grating. The downstream network learns to estimate the motion direction by reducing the systematic errors and trial-to-trial variability in the readout. **b,** Validation loss reduces over training epochs. **c,** Correlation coefficients between the trained weights $W_{in}$ projected from sensory neurons to hidden neurons. Indices 1–527 correspond to intensity detectors, while the rest denote change detectors. **d,** Visualization of the trained linear decoder $W_{out}$, where the $x$ and $y$ coordinates of each dot represent the readout weight. **e,** Normalized readouts for different motion directions at various contrast levels ($c \in$ [0.25, 0.5, 0.75]). Gray arrows indicate ground truths, while colored dots and ellipses show readout mean and covariance, respectively. **f,** Systematic error (blue) and inverse trial-to-trial variability (orange) in the readout as a function of contrast. **g-h,** Tuning curves for the mean firing rate and Fano factor of hidden neurons with respect to preferred motion directions (color-coded as in **e**). Opacity corresponds to contrast levels.

The intensity of a moving grating stimulus is described by a sinusoidal function

$$I(\mathbf{x}, t) = 1 + c \cos(\mathbf{k} \cdot \mathbf{x} - \omega t), \tag{7}$$

where $c \in [0, 1]$ is the contrast, $\omega$ is the temporal angular frequency, and $\mathbf{k}$ is the spatial wave vector, indicating the motion direction. The length of $\mathbf{k}$, denoted as $k = |\mathbf{k}|$, represents the grating's spatial frequency. We model the response of sensory neurons as an inhomogeneous Poisson process with a rate function $\lambda_i(t) = \alpha I(x_i, t)$, where $x_i \in \mathbb{R}^2$ is the spatial location of the neuron and $\alpha$ acts as a gain factor, modulating the neuron's response to the stimulus intensity. Assuming the rate function varies slowly relative to the observation window $\Delta t$, the spike count $n_i$ within this interval has statistical moments given by:

$$\mathbb{E}[n_i] = \alpha \Delta t, \tag{8}$$

$$\mathrm{Cov}[n_i, n_j] = \alpha \delta_{ij} \Delta t + \tfrac{1}{2} \alpha^2 c^2 \cos[\mathbf{k} \cdot (\mathbf{x}_i - \mathbf{x}_j)] \Delta t^2. \tag{9}$$

Here, the mean intensity encodes global luminance, while contrast and spatial direction are captured in the covariance, with the spatial direction being fully encoded by the correlation coefficients. However, intensity encoding alone cannot distinguish between opposite motion directions $\mathbf{k}$ and $-\mathbf{k}$, as they yield identical covariance values. To address this, we introduce an additional input channel based on the change rate of intensity

$$\partial_t I(\mathbf{x}, t) = c\omega \sin(\mathbf{k} \cdot \mathbf{x} - \omega t). \tag{10}$$

This change detection is also modeled as an inhomogeneous Poisson process with a rate function $\beta[1 - \partial_t I(x_k, t)]$, where $x_k$ are the spatial locations of the change detectors and $\beta$ serves as the gain factor for the neural response to intensity change rate. The statistical moments for these change detectors over $\Delta t$ are

$$\mathbb{E}[n_k] = \beta \Delta t, \tag{11}$$

$$\mathrm{Cov}[n_k, n_l] = \tfrac{1}{2} \beta^2 c^2 \omega^2 \cos[\mathbf{k} \cdot (\mathbf{x}_k - \mathbf{x}_l)] \Delta t^2 + \beta \delta_{kl} \Delta t. \tag{12}$$

Notably, the variance now incorporates information about the temporal frequency $\omega$. New information about motion direction emerges in the correlation between intensity and its rate of change:

$$\mathrm{Cov}[n_i, n_k] = -\mathrm{Cov}[n_k, n_i] = \tfrac{1}{2} \alpha\beta c^2 \omega \sin[\mathbf{k} \cdot (\mathbf{x}_i - \mathbf{x}_k)] \Delta t^2. \tag{13}$$

Importantly, this representation of motion direction in the covariance is independent of the initial spatial phase of the intensity and change detectors, underscoring the robustness and phase-invariance of the encoding scheme.

To specify the input layer of our neural network, we consider a hexagonal grid with $N$ sites, where each site hosts one intensity detector $i$ and one change detector $k$. The inputs to the MNN can therefore be compacted in matrix form as

$$\mu = \tfrac{1}{\Delta t} \left( \mathbb{E}[n_i], \mathbb{E}[n_k] \right)^T \tag{14}$$

and

$$\Sigma = \frac{1}{\Delta t} \begin{pmatrix} \mathrm{Cov}[n_i, n_j], \mathrm{Cov}[n_i, n_l] \\ \mathrm{Cov}[n_k, n_j], \mathrm{Cov}[n_k, n_l] \end{pmatrix}. \tag{15}$$

Note that our method also works if the input neurons are scattered randomly in space.

Before applying this input to MNN, we must consider a caveat. MNN defines covariance for stationary processes in the infinite time window limit, while the covariance in Eq (3) is computed for point processes riding on slowly varying rates over a finite time window. This violates both the stationarity and the limit conditions, potentially causing deviations between the two definitions of covariance. To reconcile this, we propose preserving stationarity by ensuring that the rate varies slowly within one observation time window and collecting a sufficiently large number of spikes over the finite time window to approximate the theoretical limit. Although this encoding scheme is a simplified model, it serves as a valuable conceptual framework, highlighting the importance of covariance in neural computation. In the next section, we will demonstrate how the downstream network can be trained to decode motion directions.

## Learning to decode motion directions through covariance-based computation

Given the representation of stimuli in terms of the correlated variability of sensory neurons' responses, we can now construct a complete neural network model for detecting motion direction through a task-driven approach. To this end, we employ a recently developed computational framework for modeling correlated neural variability known as the moment neural network (MNN). The main advantage of MNN is that it faithfully captures the nonlinear coupling of correlated variability of spiking neural networks while retaining the analytical tractability of continuous rate models.

We consider a feedforward MNN consisting of a layer of sensory neurons, a single hidden layer, and a linear readout, as depicted in Fig 2**a**. The inputs are the moments of the sensory neuron responses as defined by Eqs (14) and (15). The moments of the responses of the hidden neurons are

$$(\mu', \Sigma') = \phi(W_{in}\mu + \mu_{ext}, W_{in}\Sigma W_{in}^T + \Sigma_{ext}),\tag{16}$$

where $\phi$ is the moment activation [29] [Eqs (25)–(27) in Methods] and $(\mu_{ext}, \Sigma_{ext})$ are the moments of external currents. Here, the mean external current $\mu_{ext}$ is a trainable parameter, while the covariance $\Sigma_{ext}$ is fixed at 0 mV$^2$/ms.

As the mean firing rates of sensory neurons do not reflect the motion directions and linear transformations cannot alter its information content (i.e., linear Fisher information), the nonlinear coupling through moment activation $\phi$ is crucial for extracting information about motion direction from covariance. The hidden neurons' responses are then mapped to a 2D vector representing the estimated direction (Fig 2**a**). The moments of the readout are given by

$$(\hat{\mu}, \hat{\Sigma}) = (W_{out}\mu', W_{out}\Sigma' W_{out}^T),\tag{17}$$

which describes a distribution of directional vectors (dots in the rightmost panel of Fig 2**a**), and the angle of this direction vector is the estimated motion direction $\hat{\theta}$. The network then learns to match the estimated motion direction to the ground truth, which is a unit vector with angle $\theta$ (red star).

To train the network, we introduce a loss function that targets both the average and variable discrepancies between the estimated and true directions [Eq (30)]. The network is trained on a dataset of moving gratings with directions ranging from $-\pi$ to $\pi$ for 150 epochs. As shown in Fig 2**b**, validation loss decreases with the number of training epochs and converges after about 100 epochs.

To understand the computational properties of the trained network, we first assess the structure of the model parameters. We analyze the influence of sensory neurons on hidden

neurons by calculating the column-wise correlation of synaptic weight $W_{in}$. A higher correlation for any given pair of sensory neurons suggests a more similar effect on hidden neurons. As shown in Fig 2**c**, synaptic weights are typically correlated or anticorrelated depending on the spatial position and type (intensity or change detectors) of the sensory neurons (diagonal blocks). In particular, the correlations between synaptic weights associated with intensity and change detectors are minimal (off-diagonal blocks), indicating their independent roles in information transmission. The linear decoder $W_{out}$, illustrated in Fig 2**d**, displays a ring-shaped structure, with the coordinates of each dot representing the readout weights from a hidden neuron to the readout space. This structure reflects the direction selectivity of hidden neurons that encode linearly separable information about motion direction in their mean firing rates.

We then evaluate the model performance by varying stimulus contrasts and directions. Higher contrast levels result in better alignment of the mean readout (dots in Fig 2**e**) with the ground truth (gray arrow), reducing the average discrepancy (Fig 2**f**, blue line). The readout covariances for all stimuli form a pattern (ellipse in Fig 2**e**) with principal axes parallel to the readout mean, minimizing random errors in direction estimates. We find that lower contrast values lead to wider and less eccentric covariance, increasing random errors in direction estimates. Moreover, the variance of the estimated angle $\hat{\theta}$ is approximately inversely proportional to the stimulus contrast (Fig 2**f**, orange line), aligned with the predictions of the probabilistic population code [48].

We further characterize the activity of hidden neurons by analyzing their tuning functions for mean firing rate and Fano factor. Fig 2**g** shows four representative hidden neurons with bell-shaped tuning curves. Each of these neurons has a preferred direction, and the heights of the tuning functions increase with stimulus contrast, whereas their widths remain roughly constant. We also find that the noise correlation between the activity of any pair of hidden neurons decreases with the difference in their preferred directions and is consistent with the correlation of synaptic weights projected from sensory neurons (see S2 Appendix). These features are similar to those found in direction-selective cortical neurons [43, 44] and they emerge from the model in a task-driven way without specific constraints. The tuning functions for the hidden neurons' Fano factor, resembling their mean firing rate profiles, peak at preferred directions and are amplified by stimulus contrast (Fig 2**h**). This indicates that sensory neurons encode motion direction through covariance, with higher contrast enhancing signal covariance instead of the mean, resulting in greater response variability in hidden neurons. Contrary to the notion that noisier neuronal activity hampers coding efficiency, our results show that improved readout accuracy with higher contrasts is compatible with increased variability in hidden neurons.

Overall, there are three factors determining hidden neurons' responses in Eq (16): a constant external current $\mu_{ext}$, direction-independent input statistics $\mu$ and $\Sigma_{noise}$, and direction-dependent input statistics $\Sigma_{signal}$, where $\Sigma = \Sigma_{noise} + \Sigma_{signal}$. The first two elements are not specific to direction, whereas the third, together with the trained weights, is the key to direction selectivity in hidden neurons.

## Dynamics and efficacy of covariance-based computation in SNNs

So far we have modeled the covariance computation using MNN to explicitly track the neural pairwise covariances. At first glance, this process may necessitate global knowledge of the covariance structure of the input. However, when mapped back to the corresponding spiking neural network, this is done implicitly by the stochastic process of neural spike trains, and each sensory neuron only has access to local information. To illustrate this, we simulate a

spiking neural network to show how the information hidden in the covariance structures of the input can be extracted without explicitly knowing the covariance.

Since the MNN is derived analytically from the leaky integrate-and-fire model, we can use the trained weights in the MNN to reconstruct the SNN without additional tuning. Fig 3a displays the spike trains of a representative pair of sensory neurons and that of the hidden neurons ordered according to their preferred directions. The two sensory neurons have the same spatial location and they detect light intensity and its rate of change respectively, modeled as inhomogeneous Poisson processes with oscillating firing rates (green and orange curves, left panel), as detailed earlier. Their instantaneous firing rates range from 200 to 1800 spikes per second, under the setting of stimulus contrast $c = 0.8$, stimulus gain factor $\alpha = \beta = 1$ sp/ms and the temporal angular frequency at $\omega = 1$ rad/ms. Note that the results presented below do not rely on the specific choice of gain factor and temporal angular frequency (see S3 Appendix). Hidden neurons exhibit sparser firing patterns, ranging from 0 to 200 spikes per second, as shown in the raster plot (right panel), and those with preferred directions closer to the stimulus direction display notably higher firing rates. Applying the linear decoder from the trained MNN on hidden neurons' spike trains produces direction estimates which becomes more

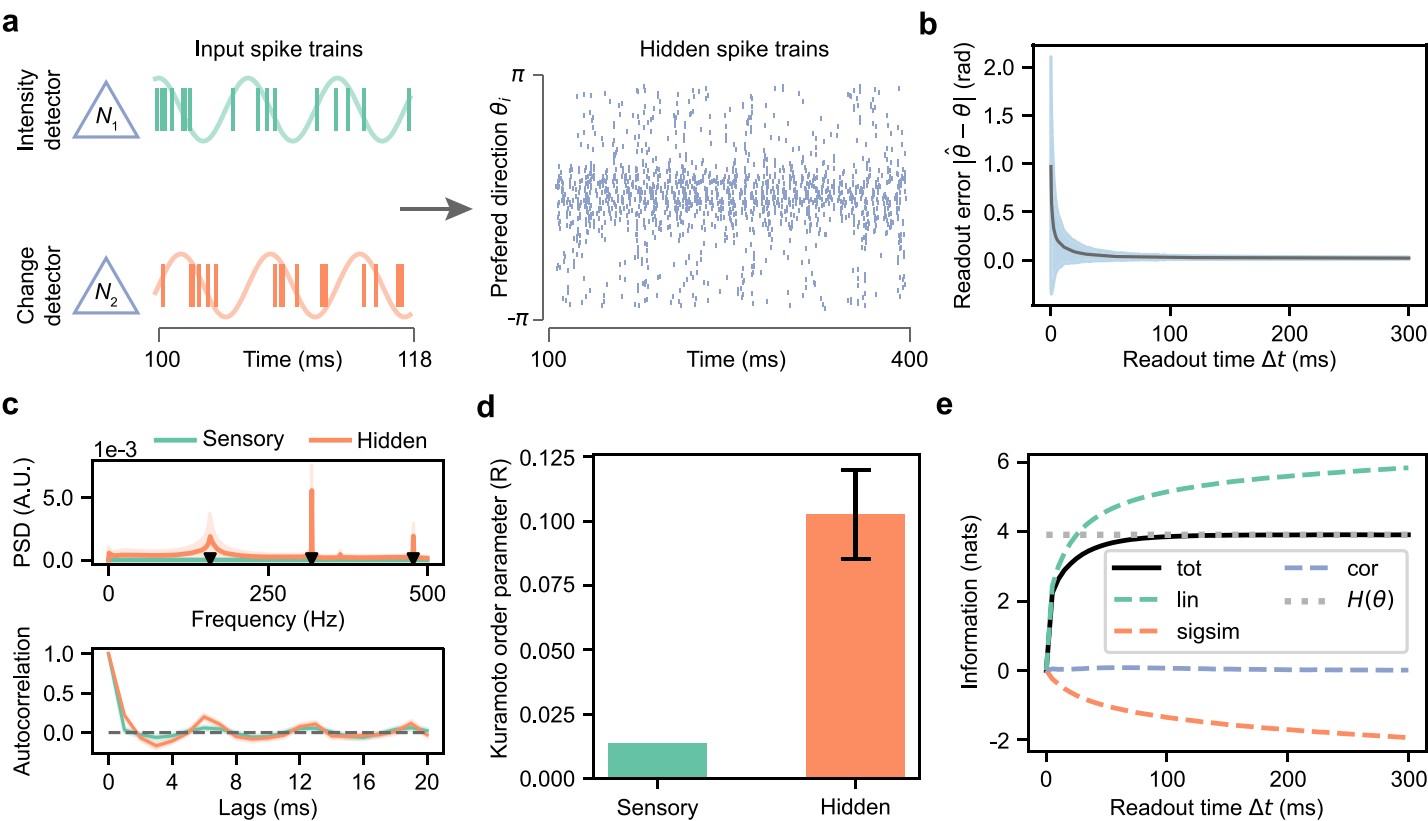

**Fig 3. The temporal dynamics of spiking neural network in inferring motion direction. a**, Spike trains of an intensity detector and a change detector at the same spatial point, along with hidden neurons of the SNN in response to a moving grating with a direction of $\theta = 0.06$ rad. The hidden neurons are organized by their preferred directions. **b**, SNN readout error as a function of readout time $\Delta t$. **c**, Power spectral density and autocorrelation of the normalized population spike count (1 ms time window). Black triangles indicate the temporal frequency of the stimulus and its integer multiples. **d**, Average Kuramoto order parameter of sensory neurons' firing rates and hidden neurons' membrane potentials from 100 to 220 ms after stimulus onset. **e**, Quantification of direction information in the readout. The gray dotted line shows the theoretical bound of mutual information, specifically the entropy of motion direction $H(\theta)$. Components: *tot*—total mutual information between stimuli and readouts; *lin*—the sum of mutual information from individual neurons; *sigsim*—redundancy due to signal similarity; *cor*—information from correlated neural activities. In **b**, **c**, solid lines and shaded regions (error bars in **d**) represent the mean and standard deviation over 500 trials, averaged across 50 motion directions.

precise with inference time. As shown in Fig 3**b**, both the mean and the variability of the readout error decrease with longer $\Delta t$ and converge to zero after about 50 ms and 100 ms, respectively.

To further understand the temporal properties of the computational process involved, we compare the power spectral density (PSD) of a normalized population spike count for the sensory and hidden layers. See Methods for how this is calculated. Fig 3**c** (upper panel) reveals that the PSD for sensory neurons is nearly zero, as oscillations in instantaneous firing rates of individual sensory neurons with different phases cancel each other out. In contrast, the PSD of hidden neurons shows distinct peaks at multiples of 159 Hz, reflecting the stimulus's temporal frequency, with peak power notably higher than those of sensory neurons. This difference is also evident in the autocorrelation of the normalized population spike count (Fig 3**c**, lower panel), which decays rapidly to zero for sensory neurons but slowly for hidden neurons, with an oscillation of about 6 ms, aligning with the temporal period of the stimulus.

Similar findings are obtained by analyzing the Kuramoto order parameter of the sensory and hidden neurons' activities (Fig 3**d**). This metric assesses the synchronization levels of the system, and they are calculated from the instantaneous firing rate for sensory neurons and the membrane potential for hidden neurons. As shown in Fig 3**d**, the Kuramoto order parameter of sensory neurons exhibits an average below 0.025, indicating asynchronous firing patterns, while hidden neurons demonstrate a higher average of approximately 0.1, signifying stronger synchrony than sensory neurons. The presence of collective oscillation in the hidden layer but not in the sensory layer suggests that the SNN is able to shift the mismatched phases of the sensory inputs. Combining with the PSD analysis above, the temporal aspect of the SNN could be a potential way to extract information about the speed of the moving grating, in addition to the motion direction. Moreover, it is worth noting that these temporal properties are not specifically targeted during MNN training and are thus obtained for free.

Next, we perform an information-theoretic analysis to quantify the amount of recovered direction information and its contributors [2, 49]. This analysis decomposes the mutual information $I_{tot}$ between the readouts and the stimulus into three components: $I_{lin}$, the information from each readout dimension separately, $I_{sigsim}$, the redundancy due to the signal similarity between readout dimensions, and $I_{cor}$, the residual information within the correlation of the readout (see Methods for details). As shown in Fig 3**e**, the mutual information $I_{tot}$ rapidly increases within the first 100 ms of stimulus presentation and eventually converges to the entropy of motion direction $H(\theta)$, indicating near-lossless transmission of direction information. The primary contributor to $I_{tot}$ is $I_{lin}$, which means that direction information is predominantly encoded in the firing rates of hidden neurons. Although $I_{lin}$ continues to rise beyond 100 ms, its growth is counterbalanced by $I_{sigsim}$, keeping $I_{tot}$ steady. This redundancy is expected because knowing one component of a direction vector limits the potential values for the other component. As the readout time windows become longer, the variance in readouts diminishes, while the mean stays approximately the same, resulting in increased redundancy. The correlation component $I_{cor}$ remains close to zero and contributes minimally to direction information. The above analysis demonstrates that the spiking neural network progressively transfers over time the information about motion direction from correlated variability of sensory neurons' responses to the mean readout.

To analyze the impact of contrast on network dynamics, we adjust the contrast within the range from 0.2 to 0.8 while keeping the gain factors at $\alpha = \beta = 1$ sp/ms and the temporal angular frequency at $\omega = 1$ rad/ms. The contrast controls the amplitude at which the instantaneous firing rate oscillates around a baseline of 1 sp/ms. In this setting, the maximum firing rate ranges from 1200 spikes per second ($c = 0.2$) to 1800 spikes per second ($c = 0.8$). It is found that a higher contrast results in a reduced readout error with quicker convergence in time and

that the readout error's trial-to-trial variability is significantly reduced with contrast, indicating a more reliable detection capability (Fig 4a). Consequently, more information can be decoded (Fig 4b). These results support our theory of covariance computation, as higher contrast enhances the strength of covariance of sensory neurons, particularly between intensity and change detectors, making it easier to decode information about motion direction.

The power spectral density of the normalized population spike count (Fig 4c) also varies with contrast. At lower contrast levels, the PSD is relatively flat. As contrast increases, the firing pattern becomes more distinct, leading to noticeable oscillations in the population spike count. These oscillations have a frequency approximately equal to the temporal frequency of the stimuli. We then plot the spike trains of the hidden neurons (Fig 4d) to illustrate how contrast affects the model behaviors, which well explained the difference observed in PSD. Although hidden neurons have different direction preferences, these preferences are weakly expressed under low contrast as the tuning functions are less sharp. As a consequence, both the task performance and the amount of decoded information are limited. As contrast increases, the firing rates of hidden neurons whose direction preference close to the presented stimulus increase, leading to a more distinct firing pattern and better task performance.

## Enhancing performance on natural image classification with covariance-based computation

We explore covariance-based computation on a more challenging task involving complex visual stimuli beyond elementary features like motion direction. For this purpose, we focus on a fine-grained classification task involving natural bird images, which includes multiple

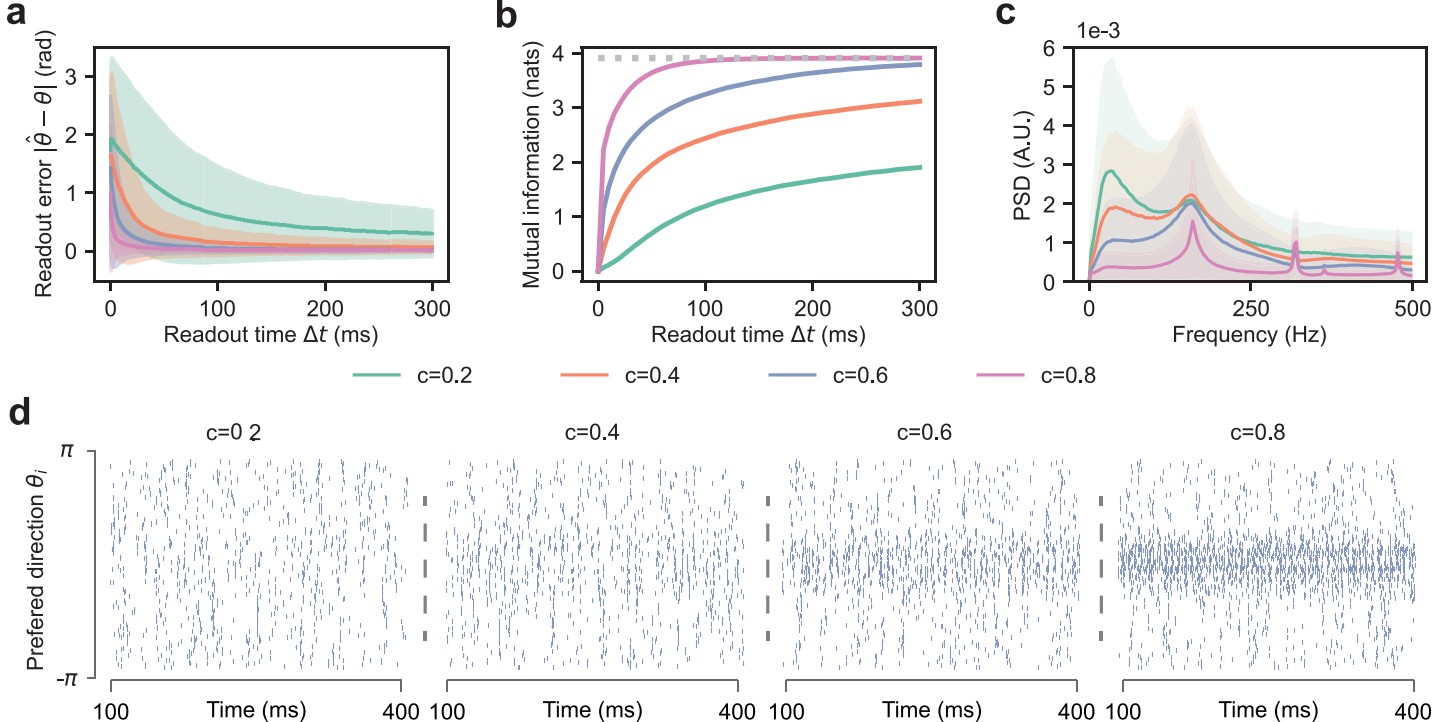

**Fig 4. The impact of contrast on SNN for detecting motion directions. a,** Readout error over time when varing the contrast. The solid line and shaded region indicate the mean and standard deviation across 500 trials, averaged over 50 motion directions. **b,** Mutual information between the cumulative readout and the motion direction of the presented stimuli under different contrasts. **c,** Power spectrum analysis of the normalized population spike count of hidden neurons under different contrasts. **d,** Raster plots of hidden neurons at different stimulus contrasts. The orientation of the presented stimulus is 0.06 radian.

subcategories within the broader category of birds [50]. The complexity of the task arises from the subtle distinctions between classes and significant intraclass variations [51]. Furthermore, this task exemplifies a perceptual disentangling challenge, as natural images possess intricate structures where the interrelations between local image patches are crucial in determining the object category.

Specifically, we consider feature maps of natural images generated by a pretrained convolutional neural network (CNN) [52], which serves as a model of visual processing in the early visual cortex [53, 54], and investigate the potential impact of feature map covariance on classification performance (Fig 5a). The CNN's output consists of a set of $c$ feature maps, each serving as a detector for a specific stimulus feature such as a bird's beak. We then flatten each spatial feature map into a temporal sequence, interpreted as the instantaneous rate $\lambda(t)$ of an inhomogeneous Poisson process. We can then calculate the mean and covariance of these feature maps of shape $c$ and $c \times c$ respectively, which are used as inputs to the downstream MNN classifier.

Before training the model, we first analyze the distribution of correlation coefficients between the CNN-derived feature maps (Fig 5b). Most correlation coefficients are found to be within the range of $(-0.1, 0.4)$ with an average of 0.0046. To assess whether these weak pairwise correlations hold significant information pertinent to image categorization, we feed the statistical moments of the CNN's feature map to a two-layer classifier. Three distinct models are considered: the first is the *correlated* model using an MNN trained with the mean and covariance of CNN's feature maps; the second is the *uncorrelated* model, using an MNN but with input correlations set to zero while retaining the variance; the third is an *ANN* model using a

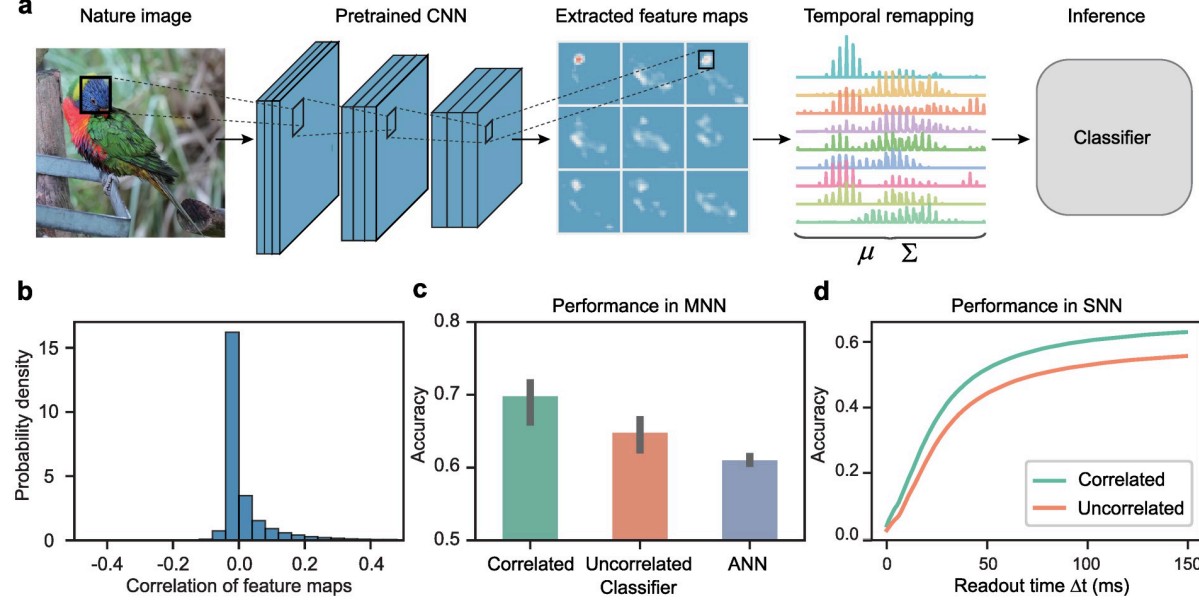

**Fig 5. Incorporating second-order information improves model performance in natural image classification. a,** Task schematic: A pretrained convolutional neural network (CNN) serves as the sensory system, extracting diverse features from the input image across multiple channels. These spatial features are then transformed into responses in the temporal domain, whose mean and covariance are computed using Eqs (2) and (3). The classifier network then utilizes the mean and covariance information to infer the image category. Note that the photograph was taken by us solely for illustrative purposes and is not in the dataset [50]. **b,** Distribution of correlation coefficients between feature maps obtained from all images in the dataset. **c,** Comparison of the classification accuracy of MNNs (*correlated* and *uncorrelated*) and an ANN after training. The error bars represent the standard deviation of 5 trials. **d,** The average accuracy of the SNNs as a function of readout time: *correlated*, an MNN trained with mean and covariance; *uncorrelated*, an MNN trained with mean and variance (all off-diagonal correlation coefficients set to zero); *ANN*, an ANN with ReLU activation trained with mean only.

rate-based artificial neural network (ANN) with rectified linear unit (ReLU) activation, trained exclusively on the mean values of the feature map. To ensure fair and accurate comparisons, we keep consistent network architectures and loss functions in these different setups. Each model is trained five times with different weight initialization.

As shown in Fig 5c, the *ANN* model trained solely on the mean of the feature maps exhibits the lowest accuracy at 60.99% on the test set. The *uncorrelated* model incorporating the variances of the feature maps as input during training shows a marginal improvement with an accuracy of 64.77%, whereas the *correlated* model incorporating the covariance of the feature maps as input shows the best improvement with an accuracy of 69.79%. Further evaluation of the temporal processing is conducted on the SNN reconstructed from the MNN models. These SNNs are given inputs that are taken from a multivariate normal distribution with the same mean and covariance as the input to the MNNs. Fig 5d illustrates that the *correlated* model consistently shows faster convergence (thus inference speed) and a higher accuracy compared to the *uncorrelated* model.

These results with natural images highlight the benefits of incorporating correlations for higher-order cortical processing beyond simple visual tasks. Despite typically weak correlation coefficients, they contain nuanced, category-specific information that complements the information provided by the mean. These findings resonate with similar observations in the field of deep learning [55, 56], further indicating the benefit of incorporating neural correlations into learning.

## Discussion

In this study, we have proposed a novel theory of covariance-based computation with spiking neurons by demonstrating how correlated neural variability can embed perceptual information and be directly involved in performing useful computation on an end-to-end basis. We introduce an encoding scheme that represents perceptual information using the covariance of sensory neurons. Through a motion direction detection task, we have shown that downstream networks can learn to extract this covariance-encoded information effectively, allowing for the transfer of perceptual information from the covariance of upstream neurons to the mean firing rates of downstream neurons. Moreover, we have demonstrated that the reconstructed SNN model could perform this computation implicitly through its temporal dynamics, achieving a nearly lossless recovery of direction information. Additionally, applying our theory to a more challenging classification task based on natural images reveals that correlated neural variability can be exploited to improve model performance and inference speed.

The notion of covariance-based computation has emerged independently across various disciplines. In cognitive neuroscience, Christoph von der Malsburg [57] introduces the idea that temporal correlation in the brain could serve as a computational resource, suggesting that the brain's information processing capability hinges on the temporal coherence of neuronal firing patterns. However, early works primarily offer conceptual explorations and do not dive into the detailed mechanisms of these processes or how they can be realized. In the realm of machine learning, it has been shown that second-order statistics among features can be exploited to improve the performance of deep neural networks [55, 58]. Although these works have shown the effectiveness of covariance in abstract models, the way biological neurons may leverage covariance for computation remains largely unknown. To address this problem, a covariance perceptron modeled after linear autoregressive processes is proposed in [40], demonstrating that the covariance structure of multivariate time series data can be transformed by the neural network to guide inference. The covariance perceptron has then been paired with a reservoir computer for solving more challenging temporal tasks such as audio processing [41,

42]. Both their works and ours place a strong emphasis on the use of covariance naturally occurring in neural activity for processing time series data, and significantly differ from conventional deep learning models for processing covariance data, which typically does not refer to any underlying stochastic process. The key difference of the covariance computation considered in our work from that in [40] is that it is modeled after spiking neural networks whose mean firing rate and firing covariability are nonlinearly coupled. As we have demonstrated, this intermingling between moments enables the information represented in the firing covariability of upstream neurons to be passed to the mean firing rate of downstream neurons in a near-lossless fashion. Our theory based on spiking neurons also has a strong biological relevance and can potentially be used to generate quantitative predictions that can be tested against experiments in humans and animals.

The covariance-based representation of stimulus features has a close connection to the concept of combinatorial code [59, 60]. A combinatorial code is a collection of activity patterns in a neural population and it is the specific combination of active neurons that encodes a particular stimulus. In particular, a combinatorial code only keeps track of what neurons are co-active and discards details about their firing rate. This is similar to the stimulus encoding proposed in this paper in that latent information about the stimulus can be entirely represented by the covariance of neural activation. For instance, in the motion direction task, each motion direction will evoke a specific pattern of firing covariability in the sensory neurons that can be used to discriminate between different stimulus directions.

In addition to this similarity, the proposed theory of covariance computation goes beyond combinatorial code by offering a biologically plausible mechanism through which a downstream neural population can effectively compute with covariance to perform useful tasks. Specifically, our theory suggests that a neural network can extract covariance-encoded information and transfer it to their mean firing rate through the nonlinear coupling of covariance. This scheme is valuable for handling noisy input, as it is concerned with the cofluctuations of neural activity over time but is insensitive to the exact values of the input. For instance, in the motion direction task, the direction is not encoded by either intensity detectors or change detectors alone but rather emerges from their temporal correlations, which cannot be fully captured by the firing rates of neurons within a single observation period. The proposed covariance neural code thus offers a concise and invariant representation of the noisy stimulus, resulting in a neural representation that remains stable even though the exact spike trains emitted can be different across trials and is insensitive to initial conditions.

In our model, downstream neurons have a twofold functionality as both integrators and coincidence detectors [61]. This is due to that the processing of perceptual information necessitates both temporal integration and cooperation among sensory neurons. We characterize this as computation based on covariance, in which the covariance of spike trains from sensory neurons influences the firing rates of downstream neurons. This indicates a potential spatiotemporal hierarchy in neural processing accomplished through repeated covariance-based computations. As one ascends this hierarchy, the spatiotemporal scale and complexity of the encoded information increase, leading to a progressively linearly separable representation of perceptual information [8, 9]. This provides a unifying perspective on the role of neural firing rate and correlation within the context of neural coding and computation. According to our theory, the perceptual information initially conveyed through the correlated activities of neurons in the early stages of processing can be progressively transformed into the mean firing rate of downstream neurons. This mechanism is strongly associated with the perceptual disentangling hypothesis [38, 39]. It suggests that the goal of brain information processing is to transform perceptual salience information, which is initially highly entangled in the input space, into a form that is linearly separable.

Covariance-based neural computation potentially offers a unique perspective on the ongoing debate about whether the brain uses rate coding or spike time coding [62, 63]. In rate coding, information is encoded by the average number of spikes over a period of time but is independent of the precise spike timing. In spike time coding, it is the precise timing of each spike that is responsible for conveying the information [64, 65]. The covariance computation instead is rooted in the principle of probabilistic coding, which lies somewhere between rate coding and spike time coding. According to our theory of covariance computation, information can be represented both in the firing rate, which corresponds to a rate code, and in the firing covariance, which summarizes certain aspects of the relative timing similar to a spike time code. A key insight is that the representation can be transformed freely between firing rate and covariance through the nonlinear coupling of signal and noise in spiking neurons. Therefore, it can be thought that rate coding and spike time coding are two sides of the same coin.

A limitation of the present study is that the MNN model is trained using backpropagation, which explicitly computes the gradients with respect to covariance. However, whether the brain uses backpropagation is debatable and it is unclear through what mechanism biological neurons can backpropagate gradients [66]. Further research could investigate in-depth the role of correlated neural variability in local learning rules and synaptic plasticity. For example, a burst-dependent synaptic plasticity rule has been suggested to modify error signals in feedback connections [67]. Such exploration could potentially explain the prevalence of noise correlation in the brain and its potential role in learning. Moreover, given that neural correlations encoding pertinent perceptual information are locally available during the forward pass in a spiking neural network, it prompts the inquiry of whether learning can occur locally with the aim of restructuring the information into the firing rate.

Another limitation of the MNN considered in our work is the assumption of stationary processes. Therefore, the correlation considered here is limited to spike count correlation over large time windows. Lag-dependent spike cross-covariance has previously been derived for a pair of neurons receiving shared input [68]. The consideration of cross-covariance could be important as two neurons that are uncorrelated at zero lag may be correlated at a different lag. However, how to generalize the MNN to nonstationary processes with cross-covariance is a nontrivial problem, both on mathematical and implementation levels. At the mathematical level, the main challenge is the derivation of a self-consistent and closed set of equations involving cross-covariance. At the implementation level, the main challenge is to develop numerical schemes that can scale up the model to a large network beyond a single pair of neurons.

With the success of deep learning in recent years, there has been a growing trend to use artificial neural network (ANN) models as a proxy to understand how information processing is done in the brain [33–35]. However, ANN models from the deep learning literature, which are originally inspired and developed from rate models for biological neural networks, neglect important aspects about how neural computation is physically implemented in the brain, where information is communicated using spiking activity with rich temporal dynamics, with neural responses on different timescales that capture different stimulus characteristics [46]. In contrast, the task-driven approach in our work employs the state-of-the-art technique to model stochastic dynamics in biological neural networks and reconnects machine learning with neurobiology [29, 30]. A broader implication for deep learning in spiking neural networks is that, by unfolding the covariance over time, an SNN is able to implicitly process the covariance mapping without requiring its global knowledge and avoids the computational cost associated with the quadratic scaling of covariance matrices [55]. Ultimately, our work paves the way for a new task-driven paradigm for studying the roles of noise and neural correlations

in the brain as well as for developing computational and learning algorithms for SNNs and neuromorphic computing.

## Methods

### Leaky integrate-and-fire neuron model

We employ the leaky integrate-and-fire (LIF) spiking neuron model

$$\frac{dV_i}{dt} = -LV_i(t) + I_i(t), \tag{18}$$

where the sub-threshold membrane potential $V_i(t)$ of a neuron $i$ is driven by the total synaptic current $I_i(t)$ and $L = 0.05$ ms$^{-1}$ is the leak conductance. When the membrane potential $V_i(t)$ exceeds a threshold $V_{\text{th}} = 20$ mV a spike is emitted, as represented by a Dirac delta function. Afterward, the membrane potential $V_i(t)$ is reset to the resting potential $V_{\text{res}}$ 0 mV, followed by a refractory period $T_{\text{ref}} = 5$ ms. The synaptic current takes the form

$$I_i(t) = \sum_j w_{ij} S_j(t) + I_i^{\text{ext}}(t), \tag{19}$$

where $S_j(t) = \sum_k \delta(t - t_j^k)$ represents the spike train generated by presynaptic neurons.

A final output **y** is readout from the spike count $\mathbf{n}(\Delta t)$ of a population of spiking neurons over a time window of duration $\Delta t$ as follows

$$y_i(\Delta t) = \frac{1}{\Delta t} \sum_j w_{ij} n_j(\Delta t) + \beta_i, \tag{20}$$

where $w_{ij}$ and $\beta_i$ are the weights and biases of the readout, respectively. A key characteristic of the readout is that its variance decreases as the time window $\Delta t$ increases.

### Moment neural network

The moment embedding approach [27, 28, 30] begins with mapping the fluctuating activity of spiking neurons to their respective first- and second-order moments

$$\mu_i = \lim_{\Delta t \to \infty} \frac{\mathbb{E}[n_i(\Delta t)]}{\Delta t}, \tag{21}$$

and

$$\Sigma_{ij} = \lim_{\Delta t \to \infty} \frac{\text{Cov}[n_i(\Delta t), n_j(\Delta t)]}{\Delta t}, \tag{22}$$

where $n_i(\Delta t)$ is the spike count of neuron $i$ over a time window $\Delta t$. In practice, the limit of $\Delta t \to \infty$ is interpreted as a sufficiently large time window relative to the timescale of neural fluctuations. We refer to the moments $\mu_i$ and $\Sigma_{ij}$ as the mean firing rate and the firing covariability in the context of MNN, respectively.

For the LIF neuron model [Eq (18)], the statistical moments of the synaptic current are equal to [27, 28]

$$\begin{cases} \bar{\mu}_i = \sum_k w_{ik}\mu_k + \bar{\mu}_i^{\text{ext}}, & (23) \\ \bar{\Sigma}_{ij} = \sum_{kl} w_{ik} C_{kl} w_{jl} + \bar{C}_{ij}^{\text{ext}}, & (24) \end{cases}$$

where $w_{ik}$ is the synaptic weight and $\bar{\mu}_i^{\text{ext}}$ and $\bar{\Sigma}_{ij}^{\text{ext}}$ are the mean and covariance of an external

current, respectively. Note that from Eq (24), it becomes evident that the synaptic current is correlated even if the presynaptic spike trains are not. Next, the first- and second-order moments of the synaptic current are mapped to that of the spiking activity of the post-synaptic neurons. For the LIF neuron model, this mapping can be obtained in closed form through a mathematical technique known as the diffusion approximation [27, 28] as

$$\begin{cases} \mu_i = \phi_\mu(\bar{\mu}_i, \bar{\sigma}_i), & (25) \\ \sigma_i = \phi_\sigma(\bar{\mu}_i, \bar{\sigma}_i), & (26) \\ \rho_{ij} = \chi(\bar{\mu}_i, \bar{\sigma}_i)\chi\left(\bar{\mu}_j, \bar{\sigma}_j\right)\bar{\rho}_{ij}, & (27) \end{cases}$$

where the correlation coefficient $\rho_{ij}$ is related to the covariance as $\Sigma_{ij} = \sigma_i\sigma_j\rho_{ij}$. The mapping given by Eqs (25)–(27) is called the moment activation, which is differentiable so that gradient-based learning algorithms can be implemented and the learning framework is known as the moment neural network (MNN) [30].

The functions $\phi_\mu$ and $\phi_\sigma$ are derived from the LIF neuron model through a Fokker-Planck formalism and they together map the mean and variance of the input current to that of the output spikes according to [27, 28, 69]

$$\begin{cases} \mu = \dfrac{1}{T_{\mathrm{ref}} + \frac{2}{L}\int_{I_{\mathrm{lb}}}^{I_{\mathrm{ub}}} g(x)dx}, & (28) \\ \sigma^2 = \frac{8}{L^2}\mu^3\int_{I_{\mathrm{lb}}}^{I_{\mathrm{ub}}} h(x)dx, & (29) \end{cases}$$

where $T_{\mathrm{ref}}$ is the refractory period with integration bounds $I_{\mathrm{ub}}(\bar{\mu}, \bar{\sigma}) = \frac{V_{\mathrm{th}}}{\sqrt{L}\bar{\sigma}}\frac{L-\bar{\mu}}{L}$ and $I_{\mathrm{lb}}(\bar{\mu}, \bar{\sigma}) = \frac{V_{\mathrm{res}}}{\sqrt{L}\bar{\sigma}}\frac{L-\bar{\mu}}{L}$. The constant parameters $L$, $V_{\mathrm{res}}$, and $V_{\mathrm{th}}$ are identical to those in the LIF neuron model in Eq (18). The pair of Dawson-like functions $g(x)$ and $h(x)$ appearing in Eqs (28) and (29) are $g(x) = e^{x^2}\int_{-\infty}^{x} e^{-u^2}du$ and $h(x) = e^{x^2}\int_{-\infty}^{x} e^{-u^2}[g(u)]^2 du$. The function $\chi$, which we refer to as the linear perturbation coefficient, is equal to $\chi(\bar{\mu}, \bar{\sigma}) = \frac{\bar{\sigma}}{\sigma}\frac{\partial\mu}{\partial\bar{\mu}}$ and is derived using a linear perturbation analysis around $\bar{\rho}_{ij} = 0$ [28]. This approximation is justified because pairwise correlations between neurons in the brain are typically weak. Numerical simulations also show that the linear approximation works reasonably well for most inputs, even when the input correlation is away from zero [29]. An efficient numerical algorithm is used to evaluate moment activation and its gradients [29].

Derived from the spiking neuron model on a mathematically rigorous ground, the moment neural network faithfully captures spike count variability up to second-order statistical moments [27, 28] and can be considered as a minimalistic yet rich description of the statistical properties of pairwise neural interactions. Meanwhile, MNN retains the analytical tractability and differentiability of continuous rate models with which gradient-based learning can be performed. The network parameters trained in this way can then be used directly to recover the spiking neural network without fine-tuning of free parameters.

## Motion direction detection task

**Model setup for training.**   The method for training the MNN model in the motion direction detection task follows our previous work [30]. The network configuration includes 1054 sensory neurons, 1054 hidden neurons, and 2 readout neurons. In the input layer, 527 pairs of intensity and change detectors are evenly distributed across a unit hexagonal grid. The stimuli are sinusoidal gratings with spatial wave number $k = |\mathbf{k}| = 5\pi$, temporal angular frequency $\omega = 1$ rad/ms, and varying contrast $c$. The sensory neurons then transduce the stimuli into spike

trains using a gain factor $\alpha = \beta = 1$ sp/ms, whose moments are found using an observation time window $\Delta t = 1$ ms, following Eqs (14) and (15). The hidden layer comprises a synaptic summation layer (Eqs 23 and 24), a moment batch-normalization layer [30], and a moment activation layer (Eqs 25–27). The moment batch-normalization is a generalization of the standard batch-normalization for rate-based neural network models [70] to the second order and its role is to facilitate gradient-based learning. The moment batch-normalization layer is re-absorbed into the synaptic summation layer after training is done (see [30] for full details).

For effective training of the model parameters $\mathbf{\Phi}$, we introduce a loss function based on the readout mean $\hat{\mu}$ and covariance $\hat{\Sigma}$,

$$L(\mathbf{\Phi}) = \int p(\mathbf{y}|\mathbf{\Phi}) \arccos\left(\frac{\mathbf{y} \cdot \mathbf{t}}{|\mathbf{y}||\mathbf{t}|}\right) d\mathbf{y} = \mathbb{E}\left[\arccos\left(\frac{\mathbf{y} \cdot \mathbf{t}}{|\mathbf{y}||\mathbf{t}|}\right)\right], \tag{30}$$

where $\mathbf{y} \sim \mathcal{N}(\hat{\mu}, \hat{\Sigma}|\mathbf{\Phi})$ represents a Gaussian-distributed readout and $\mathbf{t}$ is the direction vector of the ground truth $(\cos\theta, \sin\theta)$. To calculate the loss function, we first generate random samples $\mathbf{y} = L\mathbf{z} + \mu$, where $\mathbf{z}$ represents uncorrelated unit normal random variables and $L$ is the Cholesky decomposition $\Sigma = LL^T$, and then calculate the expected value of the loss using these samples.

The model was trained with backpropagation implemented in PyTorch for 150 epochs, using the AdamW optimizer with default learning rate (0.001) and weight decay (0.01). During each epoch, 10,000 stimulus samples were created by randomly selecting contrasts $c$ and motion directions $\theta$ from uniform distributions over $[0, 0.8]$ and $[-\pi, \pi)$, respectively.

**Model setup for SNN simulation.** The trained MNN parameters were directly used to reconstruct the SNN, as detailed in our previous work [30]. For the motion direction detection task, we maintained a consistent contrast level $c = 0.8$ and selected 50 motion directions uniformly distributed between $-\pi$ and $\pi$. The membrane potentials of hidden neurons were initially set to random values between the resting potential $V_{\text{res}}$ and the threshold $V_{\text{th}}$. To ensure accurate calculation of mean spike rates and trial-to-trial covariance for decoding, we repeated the simulations over 500 trials for each direction, each for a duration of 1256 ms at a time increment of $\delta t = 0.1$ ms. The input spikes were generated as inhomogeneous Poisson processes as described in the main text.

**Population activity analysis.** We started by calculating the population spike count $R(t)$ for the sensory and hidden neurons in 1 ms time bins, which were then centered and normalized to obtain

$$D(t) = \frac{R(t) - \langle R \rangle}{\langle R \rangle}, \tag{31}$$

where $\langle R \rangle$ was the within-trial averaged population spike count. We then computed the power spectral density $P(f) = \frac{1}{T}|\hat{D}(f)|^2$ and autocorrelation $C(\tau) = \langle D(t), D(t + \tau) \rangle$ of the normalized population spike count (as shown in Fig 3c), averaged across 50 directions and 500 independent trials.

Phase synchrony was analyzed using the firing rate of sensory neurons and the membrane potentials of hidden neurons recorded from 100 ms to 220 ms. The Hilbert transform was applied to the centered time series data to calculate the instantaneous phase $\psi_i(t)$ of each neuron. The average phase $\bar{\psi}(t)$ and the instantaneous Kuramoto order parameter $r(t)$ were determined as follows

$$r(t)e^{i\bar{\psi}(t)} = \frac{1}{N}\sum_{j=1}^{N} e^{i\psi_j(t)}. \tag{32}$$

The average of $r(t)$ over time was taken as a measure of synchrony. The mean and standard deviation of the average Kuramoto order parameter were first estimated over 500 trials for each stimulus direction and then averaged over all 50 directions.

**Information-theoretic analysis.** The mutual information between the motion direction $\theta$ and the readout $\mathbf{y}$ is defined in terms of the entropy of the readout over all stimuli and the conditional entropy for a given stimulus as

$$I_{\text{tot}}(\mathbf{y}; \theta) = h(\mathbf{y}) - h(\mathbf{y}|\theta). \tag{33}$$

The entropy $h(\mathbf{y})$ and the conditional entropy $h(\mathbf{y}|\theta)$ are given by

$$h(\mathbf{y}) = -\int p(\mathbf{y}) \log p(\mathbf{y}) d\mathbf{y}, \tag{34}$$

$$h(\mathbf{y}|\theta) = -\sum_i p(\theta_i) \int p(\mathbf{y}|\theta_i) \log p(\mathbf{y}|\theta_i) d\mathbf{y}, \tag{35}$$

where the readout distribution $p(\mathbf{y}|\theta)$ is assumed to be a Gaussian distribution with mean $\hat{\mu}$ and covariance $\hat{\Sigma}$ conditioned on the stimulus $\theta$. The mean $\hat{\mu}$ and covariance $\hat{\Sigma}$ of the readout $\mathbf{y}$ within a readout time $\Delta t$ are estimated from SNN simulations. Assuming a uniform prior and discretizing motion directions into 50 bins (matching the number of directions used for SNN simulations), the readout distribution over all stimuli is calculated as

$$p(\mathbf{y}) = \sum_i p(\theta_i) p(\mathbf{y}|\theta_i). \tag{36}$$

We use the information breakdown analysis [2, 49] to further dissect the mutual information $I_{\text{tot}}$ into three components, allowing us to assess the amount of contributions from individual readout components $I_{\text{lin}}$, signal similarity among readout components $I_{\text{sigsim}}$, and the noise correlation in the readouts $I_{\text{cor}}$. The quantity $I_{\text{lin}}$ measures the total amount of information that would be transmitted if all readout components were independent, which is given by

$$I_{\text{lin}} = \sum_j [h(y_j) - h(y_j|\theta)], \tag{37}$$

where $y_j$ is the $j$-th component of the readout. The quantity $I_{\text{sigsim}}$ measures the information loss arising from the redundancy due to overlaps between the tuning curves of each readout component, which is given by

$$I_{\text{sigsim}} = h(\mathbf{y}_{\text{ind}}) - \sum_j h(y_j), \tag{38}$$

where the independent population response $\mathbf{y}_{\text{ind}}$ is defined by the distribution

$$p(\mathbf{y}_{\text{ind}}|\theta) = \prod_j p(y_j|\theta). \tag{39}$$

The last component $I_{\text{cor}}$ accounts for the rest part of $I_{\text{tot}}$, that is, the total amount of information due to noise correlations in the readout

$$I_{\text{cor}} = I_{\text{tot}} - I_{\text{lin}} - I_{\text{sigsim}}. \tag{40}$$

Since there is no analytical expression for the entropy of unconditional readout $p(\mathbf{x})$, we estimate this information through Monte Carlo sampling. The mean $\hat{\mu}$ and covariance $\hat{\Sigma}$ are computed based on readouts over a time window $\Delta t$ for all directions (Fig 3e). Subsequently, we

generate 10000 samples of $\mathbf{x}$ based on these $\hat{\mu}$ and $\hat{\Sigma}$ to empirically estimate the information as a function of the readout time window $\Delta t$. The theoretical bound of $I_{\text{tot}}$ is the entropy of motion direction $H(\theta) = -\sum_i p(\theta_i) \log p(\theta_i) = \log 50$ nats, and we define the time of convergence using the criterion $|I_{\text{tot}} - H(\theta)| < 0.1$ nats.

### Fine-grained image classification task

**Model setup for training.** For the fine-grained classification task, we utilized the Caltech-UCSD Birds-200-2011 dataset [50], which comprises 11,788 images of 200 bird species. Adhering to the recommended dataset split, we divided it into a training set of 5,994 images and a test set of 5,794 images.

An image preprocessing was applied following previous works [55]. We resized the images to have a shorter side of 448 pixels and then center-cropped them to a size of $448 \times 448$. To augment the training data, the images were randomly flipped horizontally. The images were standardized with the RGB channel means (0.485, 0.456, 0.406) and standard deviations (0.229, 0.224, 0.225) derived from the ImageNet dataset [52].

Feature maps were extracted using a pretrained VGG16 model [52] from PyTorch. These feature maps had the shape of $(c, h, d)$ where $c = 512$ was the number of feature maps (channels) and $h = d = 28$ were the height and width of each feature map, respectively. We then flatten these feature maps to $c$ temporal sequences of length $h * d$, and the mean and covariance of these temporal sequences, with shapes $c$ and $(c, c)$, were used as inputs to the MNN classifiers.

To evaluate our theory, we compared models trained under modified versions of the inputs. The *correlated* model received the mean and covariance of the feature maps as inputs; the *uncorrelated* model had the off-diagonal entries of its covariance matrix set to zero; and the *ANN* model was trained using only the mean as input. All three models consisted of an input layer (512 neurons), two hidden layers (1024 neurons each), and a readout layer (200 neurons). These models were trained for 150 epochs over 5 repeated trials, using the standard cross-entropy loss and AdamW optimizer with the default setting.

**Model setup for SNN simulation.** We reconstructed the SNNs using the parameters of the trained MNNs under *correlated* and *uncorrelated* conditions. To evaluate the performance of the SNNs on the fine-grained classification task, we simulated the SNN for each image in the test set, for a duration of 150 ms with a time increment of $\delta t = 0.1$ ms, repeated over 100 trials. The inputs to the SNN were generated by sampling from a Gaussian distribution with mean and covariance equal to those of the respective input conditions for the MNN model. The model prediction is given by the index corresponding to the entry with the highest value in the readout vector. The accuracy of the model was determined as the fraction of correct predictions for all samples in the test set, averaged across trials.

## Supporting information

**S1 Appendix. Derivation of input moments in the motion direction detection task.** The derivation of Eqs 8, 9, 11 and 12.
(PDF)

**S2 Appendix. Analysis of response correlation among hidden neurons.** The relationship between weight $W_{\text{in}}$, the preferred direction of hidden neurons and their correlation of the response.
(PDF)

**S3 Appendix. Impact of input parameters on the SNN for detecting motion direction.** We conduct a series of experiments to investigate how the gain factors $\alpha$ and $\beta$, and the temporal

angular frequency $\omega$ affect SNN behaviors.
(PDF)

**S4 Appendix. The empirical readout distribution in SNN.** Comparison of the readout distribution and the Gaussian distribution that have the same mean and covariance in the motion direction detection task.
(PDF)

## Author Contributions

**Conceptualization:** Zhichao Zhu, Yang Qi, Jianfeng Feng.

**Data curation:** Zhichao Zhu.

**Formal analysis:** Zhichao Zhu, Yang Qi.

**Funding acquisition:** Yang Qi, Wenlian Lu, Jianfeng Feng.

**Investigation:** Zhichao Zhu, Yang Qi.

**Methodology:** Zhichao Zhu, Yang Qi, Wenlian Lu, Jianfeng Feng.

**Project administration:** Jianfeng Feng.

**Resources:** Yang Qi, Jianfeng Feng.

**Software:** Zhichao Zhu, Yang Qi.

**Supervision:** Jianfeng Feng.

**Validation:** Zhichao Zhu.

**Visualization:** Zhichao Zhu.

**Writing – original draft:** Zhichao Zhu, Yang Qi.

**Writing – review & editing:** Zhichao Zhu, Yang Qi, Wenlian Lu, Jianfeng Feng.

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
