## [Decision Letter · Decision Letter 0]

23 May 2024

Dear Mr. Zhu,

Thank you very much for submitting your manuscript "Learn to integrate parts for whole through correlated neural variability" for consideration at PLOS Computational Biology.

As with all papers reviewed by the journal, your manuscript was reviewed by members of the editorial board and by several independent reviewers. In light of the reviews (below this email), we would like to invite the resubmission of a significantly-revised version that takes into account the reviewers' comments.

Both reviewers appreciate the idea that the covariance structure of the network responses can be informative about stimuli. However, reviewer 1 feels that the relation to earlier similar ideas should be explored more explicitly. Also both reviewers agree a more detailed description of the method is required.

We cannot make any decision about publication until we have seen the revised manuscript and your response to the reviewers' comments. Your revised manuscript is also likely to be sent to reviewers for further evaluation.

Sincerely,

Fleur Zeldenrust

Academic Editor

PLOS Computational Biology

Daniele Marinazzo

Section Editor

PLOS Computational Biology

Both reviewers appreciate the idea that that the covariance structure of the network responses can be informative about stimuli. However, reviewer 1 feels that the relation to earlier similar ideas should be explored more explicitly. Also both reviewers agree a more detailed description of the method is required.

Reviewer's Responses to Questions

**Comments to the Authors:**

Reviewer #1: This manuscript by Zhu et al is a very nicely written paper with a potentially powerful idea. They introduce the idea that the covariance structure of the responses of a populations of neurons can be very informative about the identity of a stimulus. It is shown that this covariance structure can be transformed into a rate code in the output layer of feed-forward spiking networks, and they showed clearly that covariance information can be important in certain cases, as illustrated in the orientation task and the bird categorization task. I would be happy to recommend it for publication once the points described below have appropriately addressed.

-My main comment is that the idea of using the covariance as way to transmit information and transform into rates is very similar to the one developed in the following 3 papers:

The covariance perceptron: A new paradigm for classification and processing of time series in recurrent neuronal networks | PLOS Computational Biology

https://journals.plos.org/ploscompbiol/article?id=10.1371/journal.pcbi.1008127

Covariance-based information processing in reservoir computing systems | bioRxiv

https://www.biorxiv.org/content/10.1101/2021.04.30.441789v2.abstract

Covariance Features Improve Low-Resource Reservoir Computing Performance in Multivariate Time Series Classification | SpringerLink

https://link.springer.com/chapter/10.1007/978-981-16-9573-5_42

In these papers the authors also show that covariance coding can solve some challenging problems, such as audio recognition. The central idea seems to be very close to the one proposed in the paper, so a thorough comparison between the previous work and the current manuscript’s goals in the introduction and discussion is needed.

-Are not spike rates used too high? In Line 275, up to 1800 spikes per second? How the performance of the spiking network degrades with lower rates?

-In Fig. 4 it is unclear how feature maps are sampled to convert the static bird image into a temporal sequence. Are these features sampled one after the other, and from them the rate is computed, from which means and covariances are finally built.

-How the analytical expression for Eqs. 24-26 are found? For LIF neurons, expressions for Eq. 24 are well-known for the case of inputs with static mean and covariance, but the expression for the correlation in Eq. 26 can be complicated:

Phys. Rev. Lett. 96, 028101 (2006) - Auto- and Crosscorrelograms for the Spike Response of Leaky Integrate-and-Fire Neurons with Slow Synapses

https://journals.aps.org/prl/abstract/10.1103/PhysRevLett.96.028101

A more detail description of the required methods and comparison between different existing methods needs to be made.

-Is a linear approximation of the output correlation being used? This seems to be only valid, according to the previous literature cited above, when LIF neurons are driven suprathreshold, but not subthreshold.

-The layers described in Lines 502-503 are not very clearly explained. How normalization is performed?

-In Line 508, output responses x do not need to be Gaussian, specially if neurons are subthreshold. Why is this approximation good here? Alternatively, one could have sampled x’s from the forward transmission of inputs to the response neurons to get a better generative model for x. Would be any difference if using the second approach?

Minor

-Line 116: I would add “we define the first and second order moments…”

-Line 246: “The first two elements”, what two elements? It might be unclear.

-The term SNN has not been defined, as far as I can see.

-In Fig. 3, y label should be phase, not phrase. Same in Line 308.

Reviewer #2: The review is uploaded as an attachment.

**Have the authors made all data and (if applicable) computational code underlying the findings in their manuscript fully available?**

Reviewer #1: Yes

Reviewer #2: Yes

PLOS authors have the option to publish the peer review history of their article (what does this mean?). If published, this will include your full peer review and any attached files.

Reviewer #1: No

Reviewer #2: **Yes: **Veronika Koren
---

## [Decision Letter · Decision Letter 1]

30 Jul 2024

Dear Mr. Zhu,

Thank you very much for submitting your manuscript "Learning to integrate parts for whole through correlated neural variability" for consideration at PLOS Computational Biology. As with all papers reviewed by the journal, your manuscript was reviewed by members of the editorial board and by several independent reviewers. The reviewers appreciated the attention to an important topic. Based on the reviews, we are likely to accept this manuscript for publication, providing that you modify the manuscript according to the review recommendations.

Both reviewers are now happy with the content of the paper. Reviewer 2 asks for a very minor revision (a supplemental figure being moved to the results section). After the authors either do this, or let me know why they would prefer not to, this manuscript would be ready for publication.

Sincerely,

Fleur Zeldenrust

Academic Editor

PLOS Computational Biology

Daniele Marinazzo

Section Editor

PLOS Computational Biology

Both reviewers are now happy with the content of the paper. Reviewer 2 asks for a very minor revision (a supplemental figure being moved to the results section). After the authors either do this, or let me know why they would prefer not to, this manuscript would be ready for publication.

Reviewer's Responses to Questions

**Comments to the Authors:**

Reviewer #1: The authors have appropriately responded to my comments.

Reviewer #2: The authors have addressed my questions thoroughly and have made a number of revisions to the text that substantially improved the clarity of the paper. The main results are now clearly presented and important ideas are suitably outlined. While some open questions remain, for example about the biological plausibility of the operating regime of the network (the firing rates of single neurons in the brain are of the order of 1-100 Hz, but in the model they seem much higher), the paper provides a number of results and concepts that will likely be useful for further study of neural coding in biological and artificial networks.

I suggest the Fig. S2 to be placed in the Results section of the main part of the paper, as it provides a comprehensive analysis of the effect of the stimulus contrast on the performance and activity of the network.

**Have the authors made all data and (if applicable) computational code underlying the findings in their manuscript fully available?**

Reviewer #1: Yes

Reviewer #2: Yes

PLOS authors have the option to publish the peer review history of their article (what does this mean?). If published, this will include your full peer review and any attached files.

Reviewer #1: No

Reviewer #2: **Yes: **Veronika Koren, Ph.D.

Figure Files:

Data Requirements:

Reproducibility:

References:

---

## [Editor Report · Decision Letter 2]

8 Aug 2024

Dear Mr. Zhu,

We are pleased to inform you that your manuscript 'Learning to integrate parts for whole through correlated neural variability' has been provisionally accepted for publication in PLOS Computational Biology.

Best regards,

Fleur Zeldenrust

Academic Editor

PLOS Computational Biology

Daniele Marinazzo

Section Editor

PLOS Computational Biology

---

## [Editor Report · Acceptance letter]

23 Aug 2024

PCOMPBIOL-D-24-00409R2 

Learning to integrate parts for whole through correlated neural variability

Dear Dr Zhu,

I am pleased to inform you that your manuscript has been formally accepted for publication in PLOS Computational Biology. Your manuscript is now with our production department and you will be notified of the publication date in due course.

With kind regards,

Anita Estes
